# ELOISE – Reliable background simulation at sub-keV energies

Holger Kluck[1]*

**1** Institut für Hochenergiephysik der Österreichischen Akademie der Wissenschaften, A-1050 Wien, Austria

* holger.kluck@oeaw.ac.at

December 27, 2022

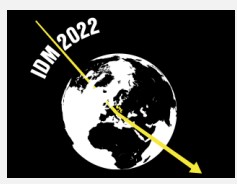

## Abstract

$CaWO_4$ and $Al_2O_3$ are well-established target materials used by experiments searching for rare events like the elastic scattering off of a hypothetical Dark Matter particle. In recent years, experiments have reached detection thresholds for nuclear recoils at the $10\,$eV-scale. At this energy scale, a reliable Monte Carlo simulation of the expected background is crucial. However, none of the publicly available general-purpose simulation packages are validated at this energy scale and for these targets. The recently started ELOISE project aims to provide reliable simulations of electromagnetic particle interactions for this use case by obtaining experimental reference data, validating the simulation code against them, and, if needed, calibrating the code to the reference data.

# 1 Introduction

The ELOISE (**R**eliable Backgr**o**und **Si**mulation at **S**ub-keV **E**nergies) project aims to validate and, if necessary, to tune and extend the Monte Carlo (MC) simulation of electromagnetic interactions in $CaWO_4$ and $Al_2O_3$ at sub-keV energies of the free particle approximation. I first motivate the necessity for a reliable simulation in these materials and at this energy scale in section 2. Afterwards, relevant physics processes for ELOISE in section 3 are identified and suitable MC codes are reviewed in section 4. In section 5, the applied methodology is outlined. The current status of ELOISE is reported in section 6 before I conclude in section 7.

# 2 Importance of a reliable background simulation

$CaWO_4$ and $Al_2O_3$ are widely used target materials for cryogenic calorimeters [1–3]. Operated at mK-temperatures, the measured phonon signal allows for detection of nuclear recoils which are the signature for highly interesting *rare* particle processes: within the Standard Model (SM) of particle physics, the Coherent Elastic Neutrino-Nucleus Scattering (CE$\nu$NS) is known to cause nuclear recoils. The NUCLEUS experiment [1] aims to measure CE$\nu$NS mainly with $CaWO_4$ calorimeters as a new probe for electro-weak precision measurements. New Physics beyond the SM may provide a particle candidate for Dark Matter (DM), one of the greatest mysteries of modern physics. The CRESST experiment [2] searches for nuclear recoils caused by the elastic scattering of hypothetical DM particles off the nuclei in $CaWO_4$ and $Al_2O_3$ among other target materials. Beyond basic research, the CRAB project [3] aims to develop a new energy calibration technique for nuclear recoils in $CaWO_4$ based on thermal neutron capturing.

In all three cases, the respective energy scales go down to the sub-keV regime: NUCLEUS and CRESST demonstrated detection threshold for nuclear recoils of 19.7 eV [4] and 30.1 eV [2], respectively. CRAB aims for a calibration signal as low as 112.5 eV [3].

At these energies, nearly all[1] particle interactions that deposits energy in the respective energy range of interest cause background events. The identification of the sought-after rare events against this dominant background relies crucially on a reliable background model.

Further interest in the detailed physics at lowest energies was sparked by the so-called *low-energy excess*: the observation by CRESST, NUCLEUS, and others of a yet unexplained increase of events with decreasing energies in the sub-keV regime, constituting an excess above the established background [6].

---

[1]For scintillating $CaWO_4$ targets, it is generally possible to separate nuclear recoils from other types of interactions via the scintillation light yield. However, at the 10 eV-scale the effectiveness of this approach degenerates quickly [2,5].

Table 1: Calculated maximum recoil energies caused by CE$\nu$NS with a neutrino of 2 MeV kinetic energy ($E_{\text{rec},\nu}$) and by elastic scattering with a $2\,\text{GeV}/c^2$-DM particle with a velocity of $220\,\text{m s}^{-1}$ ($E_{\text{rec,DM}}$) and the minimum displacement energies ($E_{\text{dis}}$) for the nuclides of CaWO$_4$ [7] and Al of Al$_2$O$_3$ [8].

|  | $_8$O | $_{13}$Al | $_{20}$Ca | $_{74}$W |
|---|---|---|---|---|
| $E_{\text{rec,DM}}$/eV | 106.4 | 69.2 | 48.6 | 11.5 |
| $E_{\text{rec},\nu}$/eV | 499.9 | 296.5 | 199.5 | 43.5 |
| $E_{\text{dis}}$/eV | 20 | 47.5 | 24 | 196 |

## 3 Relevant physics processes

Table 1 lists the maximum recoil energies caused by CE$\nu$NS or elastic DM-nucleus scattering for two typical benchmark cases. If the recoil energy ($E_{\text{rec}}$) is below the displacement energy ($E_{\text{dis}}$), as for W in the benchmark cases, it is transformed to electron and phonon excitations of the crystal lattice. In case of $E_{\text{rec}} > E_{\text{dis}}$, as is the case for O, Al, and Ca, the atom leaves its lattice site as Primary Knock-on Atom (PKA) and creates a vacancy (see e.g. [9]). In general, $E_{\text{dis}}$ depends on the orientation of the PKA relative to the crystal lattice structure [8]. Via nuclear and electronic stopping, the PKA loses kinetic energy until it stops as an interstitial atom and causes a crystal defect. At the expected PKA energies, the displaced nucleus loses its kinetic energy mainly via *Coulomb scattering* (nuclear stopping) [10]. If enough energy is transferred to secondary atoms, a nuclear recoil cascade may form. As the target crystals are operated at mK-temperatures, a recovery of the caused damage is strongly suppressed. The PKA and secondary atoms may de-excite via *fluorescence* and emission of *Auger electrons*. At a tertiary stage, the X-rays and electrons may lose their energies mainly via *photoelectric absorption* and *ionisation*, respectively (cf. fig. 1).

Since the neutron-induced background is also ultimately converted into electromagnetic interactions via the neutron-nucleus scattering and the resulting recoiling nucleus, ELOISE will be focused on the highlighted electromagnetic processes.

## 4 Suitable Monte Carlo packages

Due to the wide range of involved particles and processes as well as the complex detector geometry, the background of experiments searching for rare events is usually modelled with a general purpose MC package like Geant4 [13], FLUKA [14], or MCNP [15]. Albeit these codes were validated for various materials and energies, none of them has been validated specifically for CaWO$_4$ and Al$_2$O$_3$ at sub-keV energies. As pointed out in [16, 17], it is usually precarious to extrapolate the correctness for a given use case from studies based on a different use case. As all three codes are based on the *free particle approximation* of cold, neutral, and unbound atoms, only a dedicated validation can determine if this approximation is still suitable at the 10 eV-scale in CaWO$_4$ and Al$_2$O$_3$. Based on the displacement energies listed in table 1, I expect the onset of significant solid state-effects and hence the break-down of the free particle approximation at roughly 50 eV.

Out of the listed MC codes, the recommended application limit[2] of Geant4 is the lowest with 250 eV. Furthermore, its source code is publicly available and well documented. This has already

---

[2]See https://geant4.web.cern.ch/node/1619.

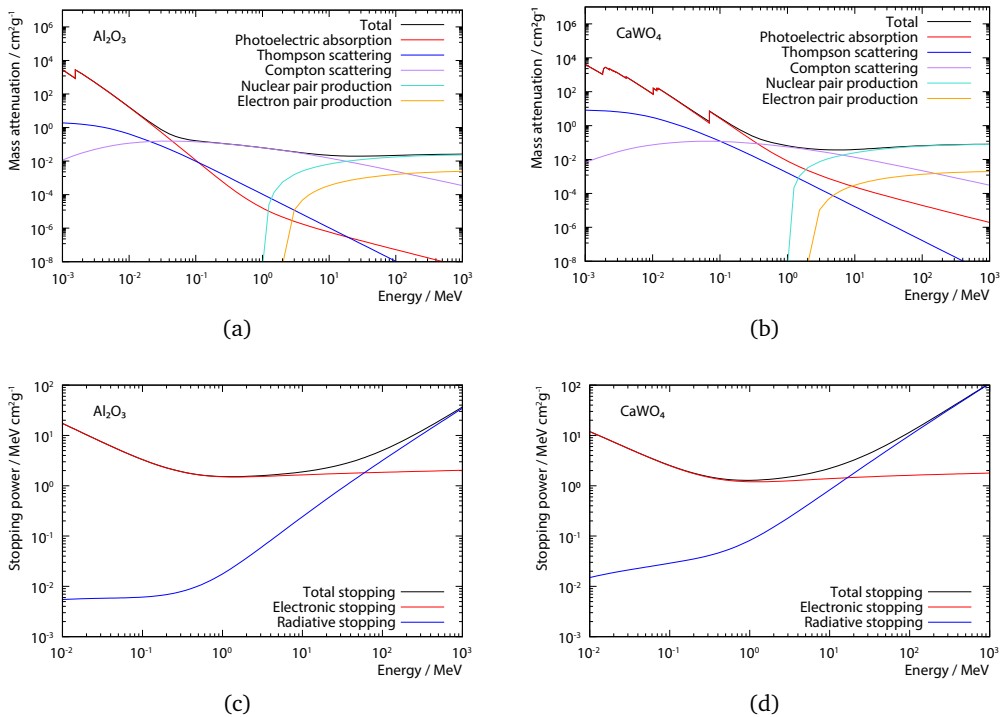

Figure 1: Literature data for attenuation of photons [11] (*top* row) and stopping power of electrons [12] (*bottom* row) in $Al_2O_3$ (*left* column) and $CaWO_4$ (*right* column).

enabled the user community to extend Geant4's applicability down to the 10 eV-scale in crystalline Si ("MicroElec" project [18]). Hence, I adopted Geant4 for my studies.

## 5 Validation protocol

To provide reliable simulations of the electromagnetic processes in $CaWO_4$ and $Al_2O_3$ from 1 keV down to the onset of solid state-effects, I will apply a four-step approach: (i) collect experimental reference data, (ii) run Geant4 based MC simulations of the reference measurements, (iii) validate the simulation against the references data, and if needed (iv) tune the existing MC models to the reference data or, if this is not sufficient, develop new models.

Step (i) will be realised either via literature searches or via dedicated measurements conducted within ELOISE. To minimise systematic errors, step (ii) requires a detailed simulation of the experimental measurement. In step (iii) the methodology developed by T. Basaglia et al. (see e.g. [17]) will be applied for the validation of Geant4 based MC simulations to provide an objective assessment. If the development of new, dedicated physics models is needed in step (iv), it is planned to adapt the approach of the MicroElec project [18].

## 6  Status of ELOISE

At the moment, I am studying the energy loss of electrons in $CaWO_4$ by ionisation. As indicated in fig. 1d, the commonly tabulated reference data goes down only to 10 keV. Hence, in step (i) a dedicated electron energy loss spectroscopy (EELS, see e.g. [19]) of a 105 nm *thin* $CaWO_4$ sample[3] was ordered. The obtained data in the energy range from 0 keV to 2 keV are presently prepared for publication. In step (ii) I am currently running a detailed simulation of the measurement based on Geant4 version 10.6.3.

## 7  Conclusion

Experiments searching at the 10 eV-scale for nuclear recoils as signature for rare events depend crucially on a reliable MC simulation of their background. At this energy scale, only electromagnetic interactions contribute to the known background. As the breakdown of the free particle approximation in $CaWO_4$ and $Al_2O_3$ can be expected at $\approx 50$ eV, a validation of the MC codes is necessary. Currently, ELOISE prepares the validation of Geant4 for electron ionisation in $CaWO_4$ against a dedicated reference measurement in the energy range of 0 keV to 2 keV.

**Funding information** The ELOISE project and HK are funded by the Austrian Science Fund (FWF): P 34778-N "ELOISE".

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
