# Peer review of "ELOISE -- Reliable background simulation at sub-keV energies"

_SciPost Physics Proceedings_

## Round 1 · Referee Report · Anonymous · 2023-1-2

Report

After the work done by the author I recommend the manuscript for publication as it is.

---

## Round 1 · Author Response

I implemented the changes that were requested by the reviewer, see also the list of changes.

---

## Round 1 · List of Changes

I am grateful to the reviewer for their helpful comments, which I copy below. I reply after the dashes "--" as following:

“CaWO4 and Al2O3 are widely used target materials for cryogenic calorimeters.” - please provide references.
-- I now refer to the CRESST, NUCLEUS and CRAB experiments to demonstrate the wide use.

“The NUCLEUS experiment [1] aims to measure CEνNS with CaWO4 calorimeters as a new probe for electro-weak precision measurements. ” and Al2O3 too judging on their last presentations
-- As discussed in [1, p.14, §6.3], the CEvNS signal is expected to be dominant only in the CaWO4 crystals, whereas the Al2O3 crystals are expected to be dominated by background. Hence, operating both targets simultaneously will provide a possibility to disentangle the signal from the background. As the preliminary usage of the Al2O3 will not be the CEvNS measurement, I leave the sentence as it is.

“Table 1 lists the maximal recoil energies for two typical benchmark cases.” Make clear it is simulations or analysis
-- The values are obtained from analytical calculations based on energy/momentum-conservation. I add a clarification to the caption.

“As the target crystals are operated at mK-temperatures, an annealing of the caused damage is strongly suppressed.” - not clear what author is trying to say here.
-- There is a non-zero probability that the caused damage to the crystal lattice is mend by thermal movement of the involved atoms, resulting in a limited recrystallization or annealing [see https://en.wikipedia.org/wiki/Annealing_(materials_science)]. However, at the operation temperatures of the crystals this movement and hence the annealing is strongly suppressed. Therefore, the caused damage is more persistent and has to be considered. I rephrased the sentence to “As the target crystals are operated at mK-temperatures, a recovery of the caused damage is strongly suppressed.”

“neutronic background” ?
-- I used “neutronic” in the meaning of “relating to, or consisting of neutrons” [see https://www.collinsdictionary.com/dictionary/english/neutronic] similar to “hadronic” or “electronic”. I replaced it with “neutron induced background”.

Fig.1 make clear is it data or simulation
-- These are data from literature. I added a clarification to the caption.

“we will apply the methodology developed by T. Basaglia et al.” please give reasoning
-- The methodology developed by T. Basaglia et al. dedicatedly aims to specify the agreement between Geant4 simulations and reference data in an objective and quantifiable way. Hence, it is perfectly suitable for the use case of ELOISE. I added, “to provide an objective assessment” to the relevant sentence to make the reason clear.

“105 nm thick CaWO4 sample” please explain why this thickness and do you plan to check other thicknesses?
-- This thickness was chosen to prevent multiple electron interactions and hence to obtain a clean sample of single electron interaction. The actual value was not determined a priori but is an outcome of the sample preparation process. A survey of different thicknesses is not planed, however the actual used thickness has to be reported to that Geant4 simulations can try to reproduce the measurement. I add a footnote to explain this.

You are currently on this page

Resubmission 2212.12634v1 on 29 December 2022

---

## Editorial Decision

accepted_in_target_journal